# Oral Mucosa Could Be an Infectious Target of SARS-CoV-2

**DOI:** 10.3390/healthcare9081068

**Published:** 2021-08-19

**Authors:** Tatsuo Okui, Yuhei Matsuda, Masaaki Karino, Katsumi Hideshima, Takahiro Kanno

**Affiliations:** Department of Oral and Maxillofacial Surgery, Shimane University Faculty of Medicine, Izumo 693-8501, Shimane, Japan; tokui@med.shimane-u.ac.jp (T.O.); yuhei@med.shimane-u.ac.jp (Y.M.); karino71@med.shimane-u.ac.jp (M.K.); hideg@med.shimane-u.ac.jp (K.H.)

**Keywords:** COVID-19, ACE2 expression in oral region, liquid-based cytology

## Abstract

The World Health Organization reported that severe acute respiratory syndrome coronavirus 2 (SARS-CoV-2) transmission is caused by respiratory droplets and aerosols from the oral cavity of infected patients. The angiotensin-converting enzyme 2 (ACE2) is considered the host functional protein for SARS-CoV-2 infection. In this article, we first revealed that the positive proportion of ACE2 expression in gingival cells collected from the gingival sulcus was increased to the same level as the tongue. Our data demonstrate that cells in the gingival sulcus may be a new entry point for the SARS-CoV-2 virus via a high expression of ACE2. In addition, we first evaluated the expression of ACE2 in various sites of the oral cavity with noninvasive, convenient liquid-based cytology. The liquid-based cytology evaluation of oral tissue may provide a novel preventive medical avenue against COVID-19.

## 1. Introduction

Coronavirus infectious disease 2019 (COVID-19) caused by severe acute respiratory syndrome coronavirus 2 (SARS-CoV-2) spread worldwide during the years 2019 to 2021 [1]. The primary entry of SARS-CoV-2 is considered to be the contact between projected droplets and cells in the oral cavity, nose or eyes [2]. Although SARS-CoV-2 can be detected in saliva, the routes of infection remain elusive, and little is known about the routes of transmission through the oral mucosa [3]. Additional clinical evidence and pathological research are thus needed to confirm the ability of SARS-CoV-2 to infect the oral tissues. A recent study of SARS-CoV-2 host cell receptor angiotensin-converting enzyme 2 (ACE2) could be valuable for the prevention and treatment of COVID-19 [4].

The infectivity of SARS-CoV-2 depends on the ability of this virus to enter the host cells, and there is clear evidence that ACE2 is the primary receptor interacting with the virus spike protein when the SASRS-CoV-2 enters cells [5,6]. The results of a recent investigation indicated that ACE2 is expressed in oral tissues and that the oral cavity can thereby be an important reservoir of SARS-CoV-2 that may serve as an entry point to the respiratory and gastrointestinal tracts [5]. The ACE2-expressing cells in oral tissues might be a direct infection route for SARS-CoV-2 [7]. However, the distribution of AEC2 in oral tissues is still unknown. Notably, olfactory and gustatory disorders are the major frequent symptoms of COVID-19.

This article provides the first report on ACE2 expression in human oral tissue, the tongue, gingiva, and palate observed by a noninvasive, convenient and common diagnostic procedure for the oral mucosal legions, i.e., oral liquid-based cytology [8]. The protocol of this study was approved by the Research Ethics Committee of Shimane University (approval no. 20200423-2) and complies with the 1964 Helsinki Declaration and its later amendments or comparable ethical standards.

## 2. Materials and Methods

In total, 20 volunteers (10 males and 10 females) were enrolled in the study, and their characteristics are shown in Table 1. The median age was 36 years. Background characteristics included age, sex, smoking and alcohol consumption.

The preparation procedure for liquid-based cytology differs from that of the conventional preparation of brush cytology [9]. In the conventional preparation, multiple exfoliated cells are spread out on multiple glass slides immediately after the collection of the cells. The advantages of liquid-based cytology are that there were fewer air-dried artefacts and less contamination with imprecise elements such as blood or debris. We obtained cells from the tongue, palate and gingival sulcus of each subject with liquid-based cytology methods. The procedure of smear collection was carried out using the Orcellex brush (Rovers Medical Devices BV., Oss, The Netherlands). The smear specimens were made in a liquid-based Pap test (Becton, Dickinson and Company, Franklin Lakes, NJ, USA). The preparation of the samples followed the instructions for SurePath™ preparations. Cells were transferred into a settling chamber on the slide where gravitational force leads to the different sedimentations of the cells. Slides were incubated with ACE2 antibodies (protein tech, Rosemont, IL, USA. 1:6000) overnight at 4 °C, followed by Histofine simple stain MAX PO (Nichirei, Tokyo, Japan.) for 20 min at room temperature as a secondary antibody. After being stained with DAB, nuclei were counterstained with Mayer’s Hematoxylin Solution. All cells were classified as either ACE2-negative or -positive cells by three experts in cytology in the Department of Pathology, Shimane University Faculty of Medicine. Human and mouse kidney tissues (US biomax #KDN242, Derwood, MD, USA) were used as positive and negative controls for antibody validation (Figure 1E–G).

All cells are classified into ACE2-positive and -negative categories by using the positive and negative control as an index. For estimating the proportion, the number of ACE2-positive cells was divided by the total number of cells. ACE2 expression was compared using the Mann–Whitney U test for each of the background factors. Age was divided into two groups according to median age, and the Brinkman index (number of cigarettes per day × years smoked) was divided into two groups according to 0 or more than 0. The significance level was set at 5%. All analyses were performed with the statistical software package IBM SPSS Statistics Version 26.0 (IBM Co., Armonk, NY, USA).

## 3. Results

Twenty participants were enrolled in this study. The median (range) of age was 36.0 (29.3–43.8) and the sex was 10 males and 10 females. We collected the medical history and any regular medicine(s); however, no one used the ACE2 inhibitor. Five volunteers (25%) were past smokers and five (25.0%) were regular drinkers (Table 1).

The ACE2 expression values were as follows: tongue, 18.2%; palate, 2.0%; and gingiva, 14.6%. (Table 2) (Figure 1B–D). The ACE2 positivity proportions in the tongue and gingiva were significantly increased compared with that in the palate. Sex, smoking and alcohol consumption were not associated with ACE2 values in the tongue, palate and gingiva (Table 1). There was no significant correlation between age, brinkman index and ACE expression in each cite (Table 3). 

In the site-specific cytology, the median (IQR) number of cells collected in the tongue was 2129.0 (1695.0–7497.0), of which the median (IQR) number of ACE2-receptor-positive cells was 478.0 (144.3–1051.0), and the median (IQR) ACE2-receptor-positive proportion in the tongue was 18.2 (8.7–25.1). The median (IQR) number of cells collected in the palate was 2597.5 (1555.8–4029.5), of which the median (IQR) number of ACE2-receptor-positive cells was 44.5 (14.5–107.5), and the median (IQR) ACE2-receptor-positive proportion in the tongue was 2.0 (0.9–2.7). The median (IQR) number of cells collected in the gingiva was 7923.5 (5457.0–14661.8), of which the median (IQR) number of ACE2-receptor-positive cells was 1323.5 (629.0–2272.3), and the median (IQR) ACE2-receptor-positive proportion in the tongue was 14.6 (7.7–20.6) (Table 2).

The analysis of variance conducted using the Friedman test showed a significant difference of *p* < 0.01, and the Wilcoxon signed-rank test as the subsequent test showed a statistically significant difference between palate and tongue and between palate and gingiva (*p* < 0.05). No statistically significant difference was found between tongue and gingiva (Figure 2).

Regarding the comparison of ACE2-receptor-positive proportions by background factors of the participants, there were no statistically significant differences in gender, alcohol consumption, or smoking history in the tongue. Comparison of the ACE2-receptor-positive proportion in the palate also showed no statistically significant difference in each factor. In addition, no statistically significant difference was observed in the comparison of ACE2-receptor-positive proportion in the gingiva (Table 3).

## 4. Discussion

COVID-19 infection causes several clinical forms ranging from dysgeusia and dysosmia to severe multiple organ failure and death. In most cases of COVID-19, dysgeusia and dysosmia may occur before pulmonary manifestations [10]. Several hypotheses can be considered potential explanations of this phenomenon. There is quantified evidence that SAES-CoV-2 affects the central and peripheral nervous systems. SARS-CoV-2 invades the central nervous system, but it rarely causes encephalitis or meningitis [11]. However, the onset of COVID-19 in almost 70% of the patients involves dysgeusia and dysosmia, and central nervous system damage is unlikely to be the source of dysgeusia and dysosmia [12]. SARS-CoV-2 invades the peripheral nerves and nociceptors, as do other viruses. Urata recently reported that the olfactory epithelium (which expresses ACE2) is sloughed off for a long time after SARS-CoV-2 infection [13]. Similar to dysosmia, the most reasonable explanation of dysgeusia may be that it is a direct effect of SARS-CoV-2 on the taste buds on the tongue. In a mouse model, ACE2 does not express at the taste buds [14]. There is little knowledge of the localization of ACE2 in human taste buds. In agreement with this, the dominant expression of ACE2 in oral tissue was reported in the salivary glands and tongue compared to the gingiva and alternative sites [15,16]. According to the first half of a 2020 study, the distribution of the SARS-CoV-2 virus in the human oral cavity was thought to be in saliva [17]. However, more recent research revealed that the distribution of SARS-CoV-2 in effusion from the gingival sulcus is significantly increased, the same as in the saliva, tongue or nasal cavity [3]. In some research, it was reported that the occurrence of oral signs and symptoms should be considered in COVID-19 patients, including tongue ulcer, candidiasis, bleeding, HSV-1 infection, geographical tongue, and thrush-like ulcers [18]. Santos indicated that ACE2 activity is stimulated by gingivitis of the periodontitis site [19]. This symptom is thought to have a relation with ACE2 expression in oral tissue and COVID-19 virus proliferation. These findings may show that dental periodontal treatment decreases the infectivity of COVID-19 virus through oral mucosa.

Interestingly, smoking history tended to increase the expression of ACE2 in oral cavity cells. However, this study was merely a preliminary evaluation with a small number of samples. A future evaluation with a larger number of samples may provide more definitive evidence that alcohol use or smoking increases the expression of ACE2 in the oral cavity. 

The present analysis first revealed that the positive proportion of ACE2 expression in gingival cells collected from the gingival sulcus was increased to the same level as the tongue. Inflammatory cytokines from periodontal pathogens in the gingival sulcus may modulate ACE2 and other SARS-CoV-2 entry proteins. Our data demonstrate that cells in the gingival sulcus may be a new entry point for the SARS-CoV-2 virus via a high expression of ACE2. In the present research, we first evaluated the expression of ACE2 in various sites of the oral cavity with noninvasive, convenient liquid-based cytology [9]. A recent study suggested that structural variations in human ACE2 may influence its binding with the SARS-CoV-2 spike protein [20]. Oral cavity cells clinically collected by noninvasive liquid-based cytology methods can be easily analyzed with other proteomic approaches, such as single-cell mass spectrometry or cryo-electron microscopy analysis [21,22].

These analyses can quantify the expression and structure of ACE2 and other possible markers, such as TMPRS2 [23,24], through all oral, head and neck areas of an entire population, and thus a liquid-based cytology evaluation of oral tissue may provide a novel preventive medical avenue against COVID-19.

The impact of the COVID-19 pandemic restricts the daily dental treatment of patients. Our present findings indicate that the inflammatory gingival sulcus is a novel shedding route of SARS-CoV-2. Special periodontal treatment by dental professionals at a dental office may decrease the expression of ACE-2 in the gingival sulcus cells, thereby preventing SARS-CoV-2 viral attachment and penetration via ACE-2 protein.

## 5. Conclusions

ACE2 expression is still not fully understood, including ACE2 expression in the human oral cavity and its variation among different races [25]. Liquid-based cytology is an easy, useful and noninvasive method to evaluate these important clinical topics. The liquid-based cytology evaluation of oral tissue may provide a novel preventive medical avenue against COVID-19. 

## Figures and Tables

**Figure 1 healthcare-09-01068-f001:**
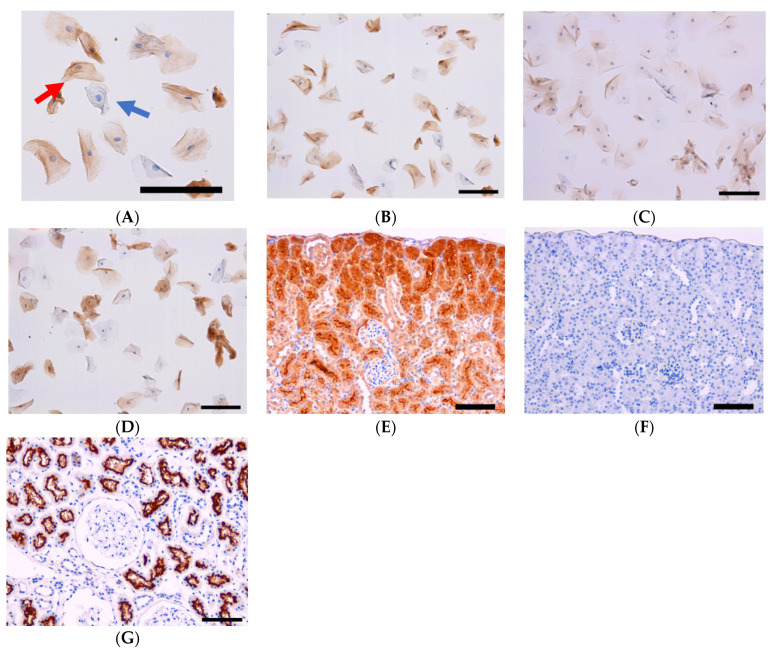
(**A**) Typical case of ACE2-positive and -negative cells. Blue arrow: ACE2-negative cell. Red arrow: ACE2-positive cell. Bar =100 μm. (**B**) Representative ACE2 expression in tongue cells (bar = 100 μm). (**C**) Representative ACE2 expression in palate cells (bar = 100 μm). (**D**) Representative ACE2 expression in gingival sulcus cells (bar = 100 μm). (**E**) Positive control using mouse kidney. (bar = 100 μm). (**F**) Negative control using mouse kidney without the 1st antibody (bar = 100 μm). (**G**) Positive control using human kidney sample (bar = 100 μm).

**Figure 2 healthcare-09-01068-f002:**
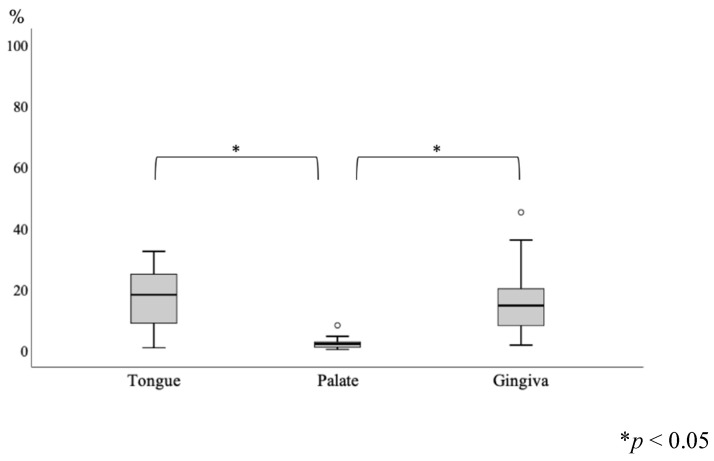
Comparison of positive proportion between groups of cytology site (n = 20). (Friedman test (*p* < 0.01) and Wilcoxon signed-rank test.)

**Table 1 healthcare-09-01068-t001:** Demographic data (n = 20).

Item	Category	N (%) or Median (Range)
Age	Total	36.0 (29–43)
	Low group (n = 9)	29.0 (26.0–30.5)
	High group (n = 11)	43.0 (38.0–55.0)
Gender	Male	10.0 (50.0%)
	Female	10.0 (50.0%)
Medical history	Hyperuricemia	1.0 (5.0%)
	Dermatitis	2.0 (10.0%)
	Hypertension	1.0 (5.0%)
	Graves’ disease	1.0 (5.0%)
	Glaucoma	1.0 (5.0%)
	Asthma	1.0 (5.0%)
Regular medicine	Benzbromarone	1.0 (5.0%)
	Minocycline	1.0 (5.0%)
	Amlodipine	1.0 (5.0%)
	Drospirenone	1.0 (5.0%)
	Mercazole	1.0 (5.0%)
	Antihistamine	1.0 (5.0%)
Alcohol consumption	Regular drinkers	5.0 (25.0%)
	None or Social drinkers	15.0 (75.0%)
Smoking history	Yes	5.0 (25.0%)
	No	15.0 (75.0%)
Brinkman Index	Total	0.0 (0.0–7.5)
	Low group (n = 15)	0.0 (0.0)
	High group (n = 5)	30.0 (14.0–375.0)

**Table 2 healthcare-09-01068-t002:** Result of cytology (n = 20).

Item	Category	Median (IQR)
Tongue	Total amount of cells	2129.0 (1695.0–7497.0)
	ACE2 receptor negative	993.0 (556.0–1864.3)
	ACE2 receptor positive	478.0 (144.3–1051.0)
	Positive proportion (%)	18.2 (8.7–25.1)
Palate	Total amount of cells	2597.5 (1555.8–4029.5)
	ACE2 receptor negative	1498.0 (879.0–2120.0)
	ACE2 receptor positive	44.5 (14.5–107.5)
	Positive proportion (%)	2.0 (0.9–2.7)
Gingiva	Total amount of cells	7923.5 (5457.0–14,661.8)
	ACE2 receptor negative	2984.0 (1208.3–4715.0)
	ACE2 receptor positive	1323.5 (629.0–2272.3)
	Positive proportion (%)	14.6 (7.7–20.6)

IQR: interquartile range.

**Table 3 healthcare-09-01068-t003:** Comparison between groups using Mann–Whitney U test (n = 20).

Item	Category	Median (IQR)	*p*-Value
ACE2-receptor-positive proportion of tongue			
Age	Low group	17.7 (11.0–29.1)	0.50
	High group	18.7 (7.7–24.5)	
Gender	Male	11.2 (6.7–23.1)	0.14
	Female	19.6 (14.5–28.2)	
Alcohol consumption	Regular drinkers	18.7 (6.8–25.3)	1.00
	None or Social drinkers	17.7 (8.6–25.4)	
Smoking history	Yes	25.4 (14.0–29.1)	0.14
	No	16.2 (7.7–22.4)	
Brinkman index	Low group	16.2 (7.7–22.4)	0.14
	High group	25.4 (14.0–29.1)	
ACE2-receptor-positive proportion of Palate			
Age	Low group	1.9 (0.9–3.6)	0.94
	High group	2.2 (0.9–2.7)	
Gender	Male	1.6 (0.9–2.2)	0.19
	Female	2.6 (1.1–3.2)	
Alcohol consumption	Regular drinkers	2.5 (2.1–3.0)	0.17
	None or Social drinkers	1.4 (0.8–2.7)	
Smoking history	Yes	2.2 (1.7–2.5)	0.80
	No	1.8 (0.8–3.2)	
Brinkman index	Low group	1.8 (0.8–3.2)	0.80
	High group	2.2 (1.7–2.5)	
ACE2-receptor-positive proportion of Gingiva			
Age	Low group	17.1 (11.8–20.1)	0.30
	High group	11.9 (3.7–22.5)	
Gender	Male	15.6 (8.1–20.1)	0.91
	Female	13.9 (6.4–23.5)	
Alcohol consumption	Regular drinkers	11.9 (6.6–38.2)	0.87
	None or Social drinkers	15.2 (8.9–19.2)	
Smoking history	Yes	21.0 (10.0–33.9)	0.23
	No	12.6 (7.3–19.2)	
Brinkman index	Low group	12.6 (7.3–19.2)	0.23
	High group	21.0 (10.0–33.9)	

## Data Availability

All data were corrected in this research. Derived data supporting the findings of this study are available from the author (T.O.) on request.

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
