# Peer review of "Oral Mucosa Could Be an Infectious Target of SARS-CoV-2"

_healthcare, 2021, doi:10.3390/healthcare9081068_

Round 1

Reviewer 1 Report

Dear Authors, 

my comments are as follows:

  1. present age with one decimal place
  2. in more places instead of  "palate" You wrote "plate"
  3. on page 3 you repeated the paragraph
  4. in Discussion part  add reference for statement: "SARS -CoV-2 invades central nervous system, but it rarely causes encephalitis or meningitis" and also for the next sentence in the text. 
  5. Explain the sentence "The impact of COVID-19pandemic restricts the daily dental treatment of patients". Do You think it is because of the patients or doctors of dental medicine? At my hospital, all procedures are performed normally. 

Author Response

We would like to thank the reviewers for their careful reading of our manuscript. We have revised the manuscript in accordance with the comments of the reviewers as described below.

We hope that we have adequately addressed the reviewers’ concerns, and that you will find this revision suitable for publication.

  1. present age with one decimal place

Thank you for pointing this out. We corrected the ages in the sentence and table.

  1. in more places instead of "palate" You wrote "plate"

We apologize for the careless mistake. We corrected to “palate” throughout.

  1. on page 3 you repeated the paragraph

We deleted the duplicate paragraph. Thank you for pointing this out.

  1. in Discussion part add reference for statement: "SARS -CoV-2 invades central nervous system, but it rarely causes encephalitis or meningitis" and also for the next sentence in the text.

We added references to these sentences, and also checked the rest of the text for appropriate citation use.

  1. Explain the sentence "The impact of COVID-19 pandemic restricts the daily dental treatment of patients". Do You think it is because of the patients or doctors of dental medicine? At my hospital, all procedures are performed normally.

We appreciate the comment from the reviewer. In Japan, a certain number of patients have avoided visiting dental hospitals due to misinformation or excessive fear of COVID-19 infection. It sounds as if the reviewer’s hospital is appropriately educating patients in regard to COVID-19. We should carry out a similar instructional campaign at our hospital.   

Reviewer 2 Report

The manuscript entitled “Oral mucosa could be an infectious target of SARS-CoV-2” describes about the angiotensin-converting enzyme 2 (ACE2) which is considered as the host functional protein for SARS-CoV-2 infection and expresses in gingival cells collected from the gingival sulcus of 10 male and female volunteers. Authors demonstrated that gingival cells might be a new entry point for SARS-CoV-2 virus by liquid-based cytology (LBC) and special oral care required by dental professional to prevent the SARS-CoV-2 infection during the dental treatment. In my opinion, the manuscript is suitable for publication in Journal ‘Healthcare’, after the authors have addressed the following concerns:

  1. Overall flow of this manuscript is good, well written and understandable.
  2. In all the tables authors can use space or horizontal lines between the two items which is representing the one set of data for the easier understanding of reader.
  3. This manuscript demonstrates the expression of ACE2 in gingival cells and its possible association with the SARS-CoV-2 virus entry by gingival sulcus. Could author explain (As shown in table 2) the possible reason for expression of ACE2 positive in some cells of tongue, palate and gingiva while some cells it was negative? Is it because of the limitation of the detection method used or because of the expression of ACE2 in few cells or if any other reason?
  4. Authors should demonstrate the expression of ACE2 between tongue, palate and gingiva cells by one more method such as western blot or ELISA.
  5. Sentence “Twenty participants were enrolled in this study…..” repeated two times need correction.
  6. Could authors explain and elaborate the line “LBC combined with other proteomic analyses can quantify the expression………….” In Discussion section.
  7. Authors should include few more lines to elaborate the conclusion.

In my opinion, the manuscript is suitable for publication in Journal ‘Healthcare’.

Author Response

We would like to thank the reviewers for their careful reading of our manuscript. We have revised the manuscript in accordance with the comments of the reviewers as described below.

We hope that we have adequately addressed the reviewers’ concerns, and that you will find this revision suitable for publication.

1. Overall flow of this manuscript is good, well written and understandable.

Thank you. We appreciate the comment.

2. In all the tables authors can use space or horizontal lines between the two items which is representing the one set of data for the easier understanding of reader.

Thank you for the suggestion. We modified the tables as you suggest.

3. This manuscript demonstrates the expression of ACE2 in gingival cells and its possible association with the SARS-CoV-2 virus entry by gingival sulcus. Could author explain (As shown in table 2) the possible reason for expression of ACE2 positive in some cells of tongue, palate and gingiva while some cells it was negative? Is it because of the limitation of the detection method used or because of the expression of ACE2 in few cells or if any other reason?

We think the expression of ACE2 in a certain number of cells in the oral cavity is negative. However, some cells activated with pathogens or other stimulants might express ACE-2 more highly than other cells. 

4. Authors should demonstrate the expression of ACE2 between tongue, palate and gingiva cells by one more method such as western blot or ELISA.

We appreciate the comment from the reviewer. Western blot analysis and ELISA analysis require a certain amount of protein. Plenty protein may not be corrected with LBC system. However, the suggestion from reviewer will be very helpful for application to future diagnoses system. We will try to carry out an in situ hybridization or single cell protein analysis in our next evaluation.

5. Sentence “Twenty participants were enrolled in this study…..” repeated two times need correction.

We apologize for this careless error. We deleted the duplicate paragraph from the manuscript.

6. Could authors explain and elaborate the line “LBC combined with other proteomic analyses can quantify the expression………….” In Discussion section.

We added several sentences to expand this point. Thank you for the suggestion.

7. Authors should include few more lines to elaborate the conclusion.

We expanded the conclusion section as well.

Reviewer 3 Report

this manuscript has merit. Could it be possible to add more references? I can only see a few articles in the Bibliography section. 

Author Response

We would like to thank the reviewers for their careful reading of our manuscript. We have revised the manuscript in accordance with the comments of the reviewers as described below.

We hope that we have adequately addressed the reviewers’ concerns, and that you will find this revision suitable for publication.

1. This manuscript has merit. Could it be possible to add more references? I can only see a few articles in the Bibliography section. 

We appreciate the comment from the reviewer. We added additional references.

Reviewer 4 Report

Dear Author(s),

Congratulations on the manuscript entitled "Oral mucosa could be an infection target for SARS-CoV-2".

It is a very important paper in light of the SARS-CoV-2 pandemic era. The role of viral particles in the mouth and the saliva needs to be clarified to establish their participation in the disease "per se" and in the pathophysiology of the oral lesions already described in those patients.

One suggestion would be regarding the patients enrolled in the study. Since most of them presented either drinking and smoking habits, it would be important to discuss the possible interference of these two characteristics on the cells studied. 

Congratulations

Author Response

We would like to thank the reviewers for their careful reading of our manuscript. We have revised the manuscript in accordance with the comments of the reviewers as described below.

We hope that we have adequately addressed the reviewers’ concerns, and that you will find this revision suitable for publication.

1. Congratulations on the manuscript entitled "Oral mucosa could be an infection target for SARS-CoV-2". It is a very important paper in light of the SARS-CoV-2 pandemic era. The role of viral particles in the mouth and the saliva needs to be clarified to establish their participation in the disease "per se" and in the pathophysiology of the oral lesions already described in those patients. One suggestion would be regarding the patients enrolled in the study. Since most of them presented either drinking and smoking habits, it would be important to discuss the possible interference of these two characteristics on the cells studied. 

We appreciate the comment from the reviewer. We added a discussion regarding the effect of smoking and alcohol consumption on ACE-2 expression in the oral cavity.

This manuscript is a resubmission of an earlier submission. The following is a list of the peer review reports and author responses from that submission.